# Unmanned-Aerial-Vehicle Data as an Effective Tool for the Evaluation of Ancient Khorasan and Modern Kabot Spring Wheat Varieties under Different Tillage Systems

Kristýna Balážová [1], Jitka Kumhálová [2,*] and Jan Chyba [1]

1 Department of Agricultural Machines, Faculty of Engineering, Czech University of Life Sciences, Kamýcká 129, 6 Suchdol, 165 21 Prague, Czech Republic; balazovak@tf.czu.cz (K.B.); chyba@tf.czu.cz (J.C.)
2 Department of Vehicles and Ground Transport, Faculty of Engineering, Czech University of Life Sciences, Kamýcká 129, 6 Suchdol, 165 21 Prague, Czech Republic
* Correspondence: kumhalova@tf.czu.cz

**Abstract:** With the changing climate, there is an increasing emphasis on drought-resistant varieties, including the ability to maintain quality production. As there is also interest in ancient wheat varieties, the aim of this study was to evaluate the growth parameters of the ancient Khorasan (Kamut®) and modern Kabot spring wheat varieties using remote sensing data. Images from unmanned aerial vehicles during four growing seasons were processed. Based on vegetation indices, the growth of these varieties and their response to meteorological conditions were evaluated, as well as the ability to resist drought and higher temperatures with respect to specific soil conditions under conventional (CT), minimum (MTC), and minimization (MTD) tillage systems. It was found that Khorasan had the lowest values of the vegetation indices on the CT variant in the dry years 2022 and 2023. On the contrary, in the previous wet years, 2020 and 2021, both varieties showed similar results. Regarding water stress, the CT variant was also the least suitable for ancient Khorasan (average Crop Water Stress Index = CWSI = 0.38). On the contrary, this variant seems to be suitable for the modern Kabot variety (CWSI = 0.29), while no significant difference between tillage variants was found for this variety. In general, water stress was easily detectable from the observed parameters in the growth phase of stem elongation ($R^2$ up to 0.88). Regarding the individual methods of tillage and water stress, the ancient variety Khorasan performed the worst with the CT variant. MTD appeared to be the best tillage method for Khorasan cultivation in terms of water management.

**Keywords:** Khorasan wheat; soil tillage; UAV; spectral indices

## 1. Introduction

Wheat is one of the world's most important commodities and has been a basic strategic food for more than 800 years [1,2]. According to grain acreage and total production volume, wheat is the second most prevalent grain worldwide [3]. Compared to the previous year, there was an increase in production in 2022/23 of about four million tons [3]. However, as the population grows, so does its consumption. The demand for plant production will increase by up to 70% by 2050, as predictions show [4]. According to Singh et al. [5], climate change and the increasing pressure of diseases and pests threaten the production of wheat and other important crops every year. Kurunc et al. [6] stated that all the mentioned stress factors significantly affect the losses of average yield by more than 50% of the agricultural production all over the world. The most important factor that influences plant production and yield is water [7]. Therefore, knowledge about the influence of water in the form of precipitation or artificial irrigation on plant growth in different growing conditions (habitat, method of tillage) is essential [6]. It is also important to choose a suitable drought-resistant variety, as pointed out by research conducted by Nakhforoosh et al. [8], where Khorasan wheat (*T. turanicum*) responded to reduced annual rainfall during the growing season

by having a later canopy. However, its high yield stability provides an underutilized genetic potential that can be a source of interesting adaptation processes for future breeding material that will have improved properties with respect to drought tolerance. Modern wheat varieties, such as the Kabot variety used in our case, are bred for high yields. Nevertheless, high inputs are only sometimes needed to achieve them, and in the case of genetic material, they are narrowly homogeneous and do not provide a wide range of different adaptive properties [9,10]. In today's changing environment, the necessary diversity of species and varieties, which store valuable genetic material, is preserved [11]. A plant is a complex organism that is constantly forced to respond to fluctuating water intake and other factors. As a result of these abiotic stresses, plant performance deteriorates, resulting in a loss of vitality and lower yield [12]. It is therefore necessary to monitor plant vitality in real time, which can be performed in an environmentally noninvasive manner using remote sensing [13].

Currently, unmanned aerial vehicles (UAVs) equipped with multispectral or thermal cameras are very useful tools for the monitoring of crop conditions in precision agriculture [14]. The data obtained from these devices can quickly and effectively provide information on the occurrence of diseases and pests [15]. However, in general, UAVs have their limitations. These mainly concern the impossibility of campaigns in bad weather, or relatively expensive equipment in the case of multispectral or thermal cameras. Virnodkar et al. [16] reported that the obtained data can also be used to predict yield or detect water stress during vegetation. The spectral properties of plants are significantly influenced by the content of chlorophyll in the plant tissue, which can be well monitored in the near-infrared band [17]. The condition of the plant canopy is most often evaluated using vegetation indices as a ratio of reflectance in two or more bands. Vegetation indices are mostly based on strong reflectance in the near-infrared part and strong absorption in the red part of the electromagnetic spectrum [18], e.g., the Normalized Difference Vegetation Index (NDVI). This vegetation index is generally used to evaluate the structure, vitality, and various biophysical processes of plants together with health status [19]. In addition, Domínguez et al. [20] stated that the advantage of the NDVI is its easy readability, as the obtained values clearly indicate the state of the canopy. In general, values below 0.25 indicate bare soil; higher values, on the other hand, indicate a different level of plant coverage and vitality in the monitored crops. The most vital plants thus have a value as close as possible to 1 [2]. On the other hand, a generally known limitation is the saturation effect, which makes it impossible to use the NDVI at its high values. The Green NDVI (GNDVI), developed by Gitelson et al. [21], uses the near-infrared and green electromagnetic parts of the spectrum in different proportions. This vegetation index is commonly used for a wide range of crops, such as garlic, grapevines, potatoes, oats, and others [22], for the purpose of chlorophyll and nitrogen content monitoring in green vegetation. Mangewa et al. [23] mentioned that this index is characterized by its high sensitivity to chlorophyll and reduces non-photosynthetic effects. In addition, it is also used for valuable and complex information for landscape evaluation [24]. Another index that enables measuring chlorophyll content is the Chlorophyll Index Red Edge (CIR), which is much more sensitive even to small changes in content in the canopy and the detection of senescent processes [25]. Water stress is an increasingly monitored parameter in plants, which is often calculated using the Crop Water Stress Index (CWSI) designed by Idso et al. [26]. The CWSI is normalized by the temperature difference between the canopy and the air by the vapor pressure deficit (VPD), which allows a comparison of the water status of the plant in a wide range of crops (vines, wheat, rice, sunflower, corn, or cotton) under different conditions [27]. After editing the index by Jones et al. [28], reference temperatures Tdry and Twet were added for a more accurate interpretation of the results. Many authors have stated the CWSI's potential for crop irrigation [29]. However, the best choice of thermal indices for accurate information on water stress is still unclear [30].

Tillage is an important part of agriculture. It is characterized by many field operations that fundamentally affect the chemical and physical properties of the soil and, thus, the

subsequent vitality, growth, and crops [31]. The most common tillage methods include conventional tillage (CT), where the basic working tool is a plow, which can turn the soil well and at the same time loosen it, which increases humus and provides mineral contributions [32]. Almagro et al. [33] warn that intensive soil cultivation results in a greater sensitivity of the soil to increasingly extensive climate changes. Melero et al. [34] showed that CT improves bulk soil mass after harvest the most. At the same time, they also pointed out that CT especially prevails in areas that have a larger amount of precipitation during the year. In some areas, for example in the Mediterranean, this method is not suitable and has a negative effect on the organic fractions of the soil and its biochemical quality [34]. Conventional tillage is the most widely used method in the Czech Republic. In 2016, 66.5% of arable land was cultivated with a plow, 32.1% with conservation technology, and 1.4% with zero tillage [2]. On the other hand, according to the Statistical Yearbook of the Environment of the Czech Republic 2021 [35], the acreage of agricultural land in organic farming and the number of ecologically managed entities are constantly growing on average by a 0.5% share of the agricultural land fund.

Minimum tillage (MTC) is based on reducing the depth or intensity of processes; at the same time, this method is characterized by a reduction in the number of individual operations [36]. In Chețan et al.'s [37] study, MTC minimum treatment (chisel treatment) in maize cultivation can be considered comparable to CT, as the results showed negligible differences between yields. The research by Chirita et al. [38] pointed to good yields in winter wheat with MTC when this variant was treated with high doses of fertilization. Minimalization tillage (MTD) with the use of a disc cultivator is another alternative method of tillage. According to Özpinar and Çay [39], MTD can ensure a higher rate of bulk density than CT during wheat cultivation. They also found that, at the same time, there were differences in hydraulic conductivity (Ks) during vegetation, when Ks was higher in MTD than in CT; on the contrary, during harvest, Ks was higher in CT than in MTD. Šíp et al. [40] stated that minimal tillage is becoming more and more popular in the Czech Republic, especially in areas with higher heat and dryness and lighter soil. Different tillage methods have their advantages and disadvantages. In general, it is important to choose one that suits the current requirements and site-specific conditions as it affects the physical properties of the soil, such as the bulk density of the soil, a parameter often used to describe levels of soil compaction [27,41].

It is clear from the previous text that ancient wheat varieties can offer several positive traits that can be used for further breeding material. They are more flexible in response to abiotic stress and often offer a richer spectrum of nutrients in their grains. However, the behavior of these ancient varieties, if they are grown using modern soil tillage technologies, has not yet been studied.

Therefore, the objective of this study is to assess Khorasan and Kabot spring wheat varieties using remote sensing data during the entire four growing seasons (from 2020 to 2023) with different meteorological conditions and to evaluate the growth of these varieties with respect to specific soil conditions under conventional, minimum, and minimalization tillage.

## 2. Materials and Methods

### 2.1. Experimental Plot Design

The study was undertaken from 2020 to 2023, four entire growth seasons, in six regular plots of 4 × 50 m near Čenovice (N 49°47′42.78″, E 15°6′35.94″) in the Central Bohemia region in the Czech Republic. The average annual temperature was between 6 and 7 degrees Celsius. The average annual precipitation was between 650 and 750 mm. The monthly temperatures and precipitation for the selected years are given in Table 1. The experimental area had a southeastern aspect with a 2.63° slope and an average elevation of 481 m a.s.l. The soil type was Cambisol, according to the World Reference Base (WRB), containing 15.1% clay, 30.9% silt, 9.2% very fine sand, 14.3% fine sand, 12.1% medium sand, 9.4% coarse sand, and 9% very coarse sand. Soil texture determination was performed on a Horiba LA-960 device (laser diffraction, dry dispersion). The soil had low levels of threat from compaction

and wind erosion. Before the plot establishment, there was long-term permanent grassland. The experimental area was divided into six plots, each 4 × 50 m (200 m²) in size, with a 4 m wide handling gap in between. First, the wheat variety Khorasan (Kamut®—*Triticum turgidum* ssp. *Turanicum*) was sown in one half of the trial area (plots 1–3) and the second wheat variety Kabot (*Triticum aestivum*) was sown in the other half (plots 4–6). The results from plots 1 to 6 were sorted according to the variants, where 1 and 4 were cultivated using conventional tillage with moldboard plowing at a depth of 20–25 cm, 2 and 5 were cultivated with minimum tillage using a coulter cultivator twice (at a depth of 15 cm), and 3 and 6 were cultivated with minimization tillage using a disc cultivator twice (at a depth of 12 cm) (see Figure 1). The machines used for soil tillage were as follows: CT—moldboard plow, manufacturer: ROSS UNIVERSUM s.r.o., type: 5–PHX–35; MTC—coulter cultivator, manufacturer: OpaLL–AGRI s.r.o., type: MERKUR II. (4 m); MTD—disc cultivator, type: BDT (4 m).

**Table 1.** Average monthly temperatures and the sum of precipitation during the growing seasons of the monitored crops.

| Month | Average Temperature (°C) | | | | Sum Precipitation (mm) | | | |
|---|---|---|---|---|---|---|---|---|
| | 2020 | 2021 | 2022 | 2023 | 2020 | 2021 | 2022 | 2023 |
| April | 9.8 | 5.1 | 6.0 | 6.2 | 17.27 | 13.71 | 39.88 | 83.82 |
| May | 11.3 | 10.2 | 14.0 | 12.4 | 166.64 | 35.3 | 43.12 | 35.78 |
| June | 15.7 | 19.8 | 18.3 | 16.5 | 206.23 | 144.53 | 86.84 | 45.77 |
| July | 18.3 | 19.4 | 18.2 | 19.4 | 136.65 | 135.36 | 58.47 | 68.27 |
| August | 18.0 | 16.7 | 18.7 | 18.5 | 148.48 | 174.76 | 117.32 | 140.37 |

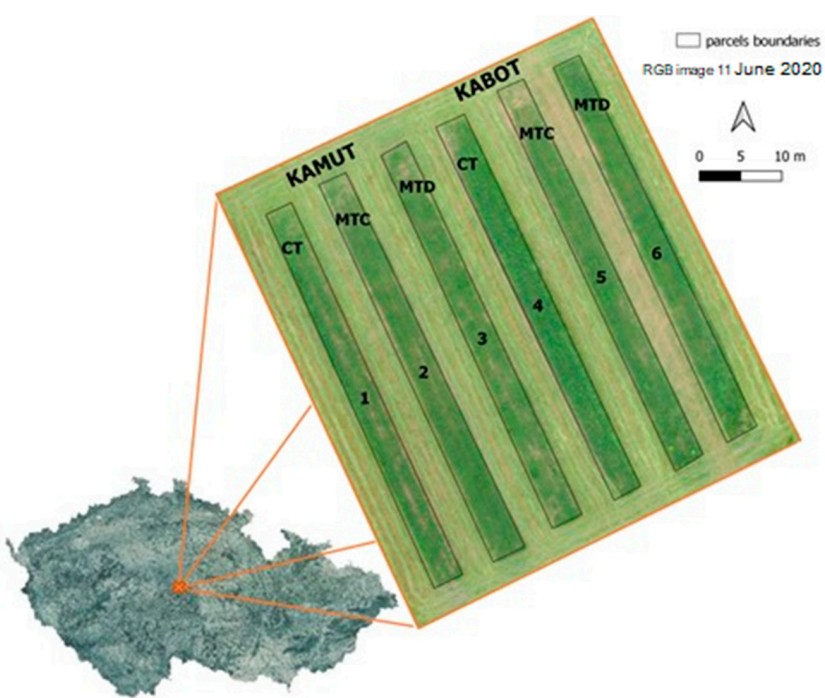

**Figure 1.** Experimental plot according to varieties and tillage variants (CT—conventional tillage; MTC—minimum tillage; and MTD—minimization tillage).

### 2.2. Agrotechnical Operations

The sowing of Khorasan and Kabot spring wheat varieties took place on 12 April 2020, 27 April 2021, 22 April 2022, and 8 May 2023. After leveling the soil surface using harrows, the NPK (at a percentage ratio of 15:15:15) fertilizer was manually scattered on the individual plots at a rate of 200 kg·ha⁻¹ (4 kg per plot). Further fertilization involving

nitrogen, phosphorus, and potassium (NPK™) took place during the tillering of crops (end of May). Herbicide Mustang™ Forte (CORTEVA™ agriscience, Indianapolis, IN, USA) was applied against the developed dicotyledonous weeds 14 days after sowing. A dose of 200 kg·ha$^{-1}$ of ammonium saltpeter with limestone (LAV) fertilizer was applied manually during the elongation growth stage. The crop was harvested using the FORTSCHRITT E516B (Fortschritt Landmaschinenwerke Neustadt/Sachsen, Germany) or New Holland CX 8080 combine harvesters (CNH Corp., Torino, Italy) for individual plots. The combine harvester hopper was emptied onto an unfolded tarpaulin, and the contents of each plot were separated and weighed.

*2.3. UAV Campaigns*

UAV campaigns corresponding to the main growth phases of spring wheat were carried out in four growing seasons from 2020 to 2023. The meteorological and phenological information at the time of scanning is given in Figure 2. A fixed-wing eBeeX drone equipped with two types of camera payloads (multispectral MicaSense RedEdge MX (AgEagle Aerial Systems Inc., Wichita, KN, USA) and thermal Duet T dual camera (senseFly SA, Route de Geneve 38, Cheseauxsur-sur-Lausanne, Switzerland)), comprising an infrared sensor developed by FLIR technology and a S.O.D.A. camera in the visible part of electromagnetic spectra for reference of the thermal sensor, was used for scanning the experimental area (detail in Table 2).

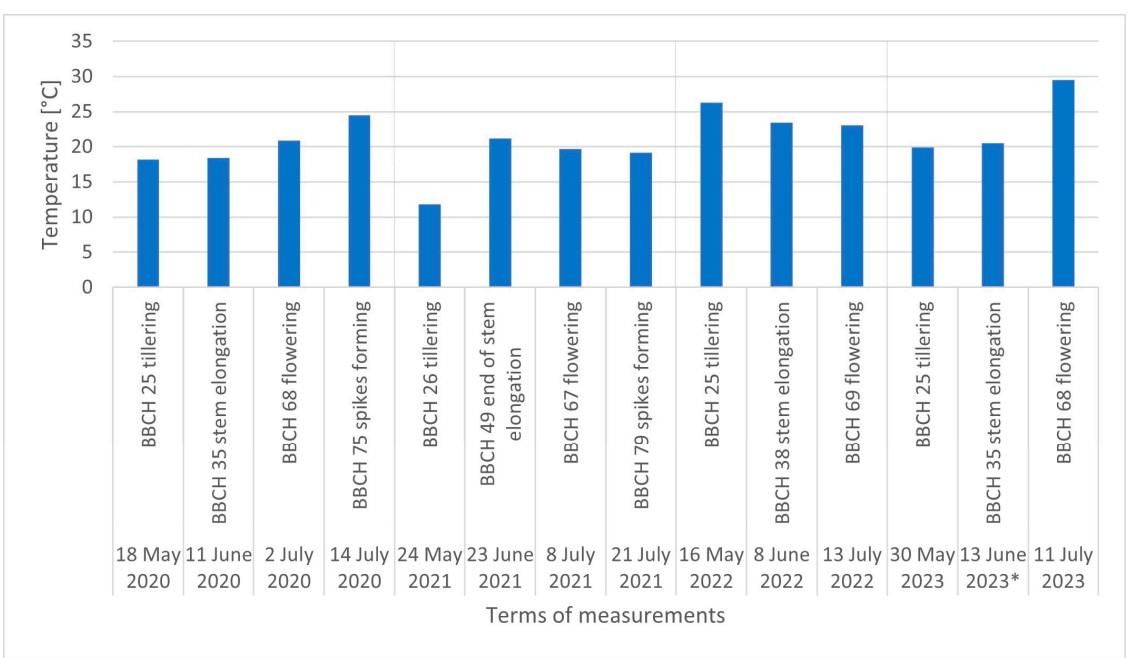

**Figure 2.** Temperature at time of measurement (°C) and growth stages at the time of scanning. * measured only with multispectral Micasense RedEdge MX camera, not with Duet T camera, due to bad meteorological conditions.

Stable conditions for the entire campaign were ensured by choosing calm days. The crop scanning was always performed at the same time around noon when the sun was in nadir. The flight parameters and postflight corrections were set in eMotion SW, following the sensor producer's recommendations, to ensure good results for deriving the final product as spectral indices or an orthophoto for data analysis. The lateral and longitudinal overlaps and flight height were as follows for the cameras used in these campaigns: MicaSense RedEdge MX camera: 75%, 75%, and 88 m above elevation data (AED); S.O.D.A. camera: 83%, 84%, and 91.8 m AED; IR sensor on Duet T camera: 75%, 80%, and 91.8 m AED.

**Table 2.** Spectral properties of sensors used in this study, and spatial resolution of the resulting images.

| Sensor/Camera | MicaSense RedEdge MX | S.O.D.A./DuetT | IR Sensor/DuetT |
|---|---|---|---|
| BLUE (nm) | 475 (20) | 450 (100) | - |
| GREEN (nm) | 560 (20) | 520 (100) | - |
| RED (nm) | 668 (10) | 660 (130) | - |
| RED EDGE (nm) | 717 (10) | - | - |
| NIR (nm) | 840 (40) | - | - |
| Thermal (μm) | - | - | 7.5–13.5 |
| Spatial resolution | 6.0 cm/px | 2.1 cm/px | 12 cm/px |

The resulting products reached an average accuracy of 4.7 cm. The images were corrected using VRS.MAX-CZEPOS (master auxiliary stations, RTCM 3.1. correction format) provided by the Czech Office for Surveying, Mapping and Cadastre of the Czech Republic.

The images were processed in Pix4D SW (Pix4D S.A., Lausanne, Switzerland), where spectral indices (details in Table 3) were calculated. The resulting indices in the form of raster data (GeoTIFF, WGS84 UTM Zone 33N coordinate system) were analyzed in QGIS SW using zonal statistics and advanced tools focused on individual parcels. A Digital Terrain Model (DTM) was derived from the orthophoto in m above sea level (a.s.l.). The data were then statistically processed in Statistica SW (TIBCO Software Inc. (2018) Statistica (data analysis software system), version 13. http://tibco.com).

**Table 3.** Spectral indices and parameters evaluated in this study.

| Spectral Index | Equation | Used for | References |
|---|---|---|---|
| Normalized Difference Vegetation Index (NDVI) | $(NIR - RED)/(NIR + RED)$ | Biomass, structure, and vigor | Rouse et al. (1974) [42] |
| Green Normalized Difference Vegetation Index (GNDVI) | $(NIR - GREEN)/(NIR + GREEN)$ | Chlorophyll | Gitelson et al. (1996) [21] |
| Chlorophyll Index Red Edge (CIR) | $(NIR/RedEdge) - 1$ | Chlorophyll | Gitelson et al. (2005) [43] |
| Crop Water Stress Index (CWSI) | $\dfrac{(Tc - Ta) - (Tcl - Ta)}{(Tcu - Ta) - (Tcl - Ta)}$ | Water stress | Katimbo et al. (2022) [44] |

Where RED, GREEN, BLUE = reflectance in visible part of electromagnetic spectra; RedEdge and NIR = reflectance in near-infrared part of electromagnetic spectra according to the MicaSense RedEdge MX sensor properties; Tc (°C) = measured canopy temperature, Ta (°C) = air temperature, Tcl (°C) = canopy temperature of well-transpiring or non-stressed crop (i.e., minimum Tc), and Tcu (°C) = the canopy temperature of a nontranspiring or severely stressed crop (i.e., maximum Tc). Terms (Tcu − Ta) and (Tcl − Ta) are referred to as upper and lower limits.

The CWSI was derived from thermal images for the individual terms of canopy scanning (see Figure 2). The calculation was performed in QGIS SW using a CWSI plugin [45]. The equation is based on the calculation of Jones et al. [46]. Since Tdry and Twet were not available, the calculation according to Irmak et al. [47] was used. They used a default value of 5 °C for workflow calculation.

The most important points of the methodological procedure are clearly visible from Table 4.

Table 4. The most important methodological points in time series. It was repeated for all 4 years of measurement.

| Experimental plot establishment (April to early May) | | | | | |
|---|---|---|---|---|---|
| **1** | **2** | **3** | **4** | **5** | **6** |
| Khorasan (Kamut®—*Triticum turgidum* ssp. *Turanicum*) | | | Kabot (*Triticum aestivum*) | | |
| Conventional tillage (CT) moldboard plow | Minimum tillage (MTC) colter cultivator | Minimization tillage (MTD) disc cultivator | Conventional tillage (CT) moldboard plow | Minimum tillage (MTC) colter cultivator | Minimization tillage (MTD) disc cultivator |
| Fertilizing by NPK™ (nitrogen, phosphorus, potassium), 200 kg·ha$^{-1}$ | | | | | |
| Herbicide application (Mustang™Forte)—14 days after sowing | | | | | |
| UAV scanning (eBee X; MicaSense RedEdge MX, Duet T camera)—tillering growth stage | | | | | |
| Fertilizing by LAV (ammonium saltpeter with limestone), 200 kg·ha$^{-1}$—stem elongation growth stage | | | | | |
| UAV scanning (eBee X; MicaSense RedEdge MX, Duet T camera)—stem elongation growth stage | | | | | |
| UAV scanning (eBee X; MicaSense RedEdge MX, Duet T camera)—flowering growth stage | | | | | |
| UAV scanning (eBee X; MicaSense RedEdge MX, Duet T camera)—spikes forming growth stage (2020, 2021) | | | | | |
| Harvesting (combine harvester Fortschritt E516B (2020, 2021); New Holland CX 8080 (2022, 2023) | | | | | |

## 3. Results and Discussion

### 3.1. Evaluation of the Crops and Variants Using Spectral Indices

The development of NDVI, GNDVI, and CIR spectral indices of Khorasan and Kabot spring wheat growing in different tillage systems for the four evaluated vegetation seasons is shown in Figure 3. Although these vegetation indices are based on different spectral bands (see Table 3), the course of the values was very similar for each index. Therefore, emphasis was placed on the properties that are evaluated; e.g., the NDVI is a measure of the vigor and structure of crops, and GNDVI and CIR are measures of chlorophyll content. It is evident that the development of the crops was dependent on weather conditions, as was described in Kumhálová et al. [48]. In general, the drier seasons 2022 and 2023 showed lower values of structural and chlorophyll parameters expressed in the calculated spectral indices. The mean values for each plot show that the development of the crops was also dependent on phenological phases. In dates corresponding to the tillering growing stage, the ancient Khorasan variety had lower values in comparison with Kabot for each index (see Figure 3). The research by Nakhforoosh et al. [8] showed that Khorasan wheat had the lowest cultivability compared to Einkorn and Zanduri wheat varieties. At the same time, its response to drought was a height reduction and loss of seed weight. However, rapid canopy development and soil cover by plants have a positive effect on reducing water evaporation from the soil and increasing competitiveness against weeds [49,50].

Regarding the tillage, the lowest values were in the variant of CT. The variants were evaluated for the whole parcels; therefore, lower values in this growth stage mean that Khorasan generally tillered later and worse, especially in the CT variant. The stem elongation growth stage was in June. In comparison with previous years, 2022 and 2023 showed lower values, which could have been caused by the low and uneven distribution of precipitation and different water ability in soil under various types of tillage. The last growth stage when the canopy was monitored was flowering and spikes forming in July. The weather conditions, especially the distribution and amount of precipitation and the air temperatures, are strongly reflected in the values of the spectral indices in this later period of growth stages. Lower values mean a lower proportion of green tissue as the crops mature, as can be seen from the CIR plot (see Figure 3c). Significantly lower values of Khorasan, the CT variant, were in the dry years of 2022 and 2023. Conversely, the previous years of 2020 and 2021, rich in water availability, showed similar values for both varieties.

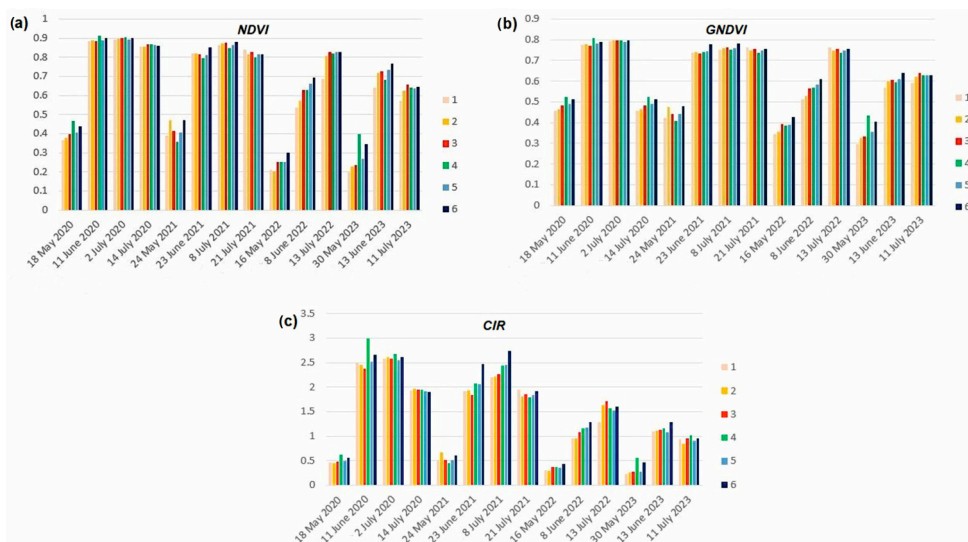

**Figure 3.** Evaluation of selected spectral indices: Normalized Difference Vegetation Index (NDVI), Green NDVI (GNDVI), and Chlorophyll Index Red Edge (CIR) for monitored seasons. 1–3: Khorasan variety; 4–6: Kabot variety; 1 and 4: conventional tillage; 2 and 5: minimum tillage; 3 and 6: minimization tillage.

In terms of tillage, the results showed that the crops achieved the best condition during these four seasons in the variants MTD and CT for the modern Kabot variety. The selected tillage could affect a whole range of physical soil properties and thus affect the growth and vitality of plants, especially in the early stages of crop development [51]. Physical properties due to tillage played a more significant role during the growing season in the drier years of 2022 and 2023. According to Woźniak and Rachoń [52], a no-tillage system had a positive effect on limiting water evaporation from a field surface. This statement agrees with Micucci and Taboada [53]. They stated in their study that plowing (CT) resulted in a greater aeration of the soil, which facilitated a faster mineralization of the organic component and thus nutrient loss.

### 3.2. Evaluation of the Crops and Variants Using the Crop Water Stress Index

The resulting values of the Crop Water Stress Index derived from thermal images for the individual terms of canopy scanning and the average for the whole period and individual monitored growing stages are given in Table 5. The CWSI values are based on the current state of the vegetation, which is influenced by meteorological variables. As shown in Table 6, the average values of the CWSI for all experimental measurements show that the Khorasan variety showed a higher level of stress than the Kabot variety. Regarding the individual methods of tillage and water stress, the ancient variety Khorasan performed the worst for the CT variant. The MTC and MTD of the tillage were very close to the CWSI values. However, MTD appeared to be the best tillage method for Khorasan cultivation in terms of water management. Likewise, the modern Kabot variety showed the best results in water management on the MTD variant. Within the monitored growth phases, it turned out that MTD was the most suitable for tillering in terms of water stress; on the contrary, the stands on the CT variant showed the greatest water stress. During stem elongation, the CSWI values were almost equal for each tillage variant in both cultivars; however, the Khorasan variety showed a greater susceptibility to water stress in this growth phase compared to the modern Kabot variety. The late phases followed the same trend. While the ancient Khorasan performed the worst on the CT variant, this variant was the most suitable for the modern variety Kabot. These results agree with Farooq's study [54], which stated that drought stress reduced biomass volume and root proliferation, fundamentally disrupted plant–water relations, and reduced water use efficiency. This trend is also supported by Kumhálová et al. [55]. The influence of topographical attributes,

e.g., flow accumulation on resulting yield, was evaluated in this study. They found that a lack of precipitation during the flowering growing stage led to water only accumulating at the lower terrain of the plot. The reason for the resulting low yield of winter wheat was a combination of soil compaction due heavy-machinery cultivation and the dry period. These led to a poor root system and bad crop vitality. Shorter plants, smaller leaf area, and conversely higher root biomass led to less damage caused by drought stress, as also mentioned in the study by Richards [56]. According to Ali et al. [57], it is beneficial to use IR-based thermal imaging because it can identify sensitive plants at the onset of drought stress or determine the stress tolerance of different cultivars. In general, plants that have cooler surfaces than others are more demanding of the water they consume and release.

**Table 5.** Crop Water Stress Index (CWSI) derived for the entire four seasons from 2020 to 2023 in monitoring dates and individual variants; average for the whole period and for the individual monitored growing stages (tillering, stem elongation, and later stages = flowering and spikes forming).

| Plots | Average | Tillering | Stem Elongation | Late Stages |
|---|---|---|---|---|
| Khorasan 1 CT | 0.38 | 0.45 | 0.30 | 0.37 |
| Khorasan 2 MTC | 0.34 | 0.37 | 0.32 | 0.33 |
| Khorasan 3 MTD | 0.32 | 0.36 | 0.31 | 0.31 |
| Kabot 1 CT | 0.29 | 0.40 | 0.23 | 0.26 |
| Kabot 2 MTC | 0.30 | 0.37 | 0.23 | 0.30 |
| Kabot 3 MTD | 0.28 | 0.32 | 0.22 | 0.28 |

| | Plots | 18 May 2020 | 11 June 2020 | 2 July 2020 | 14 July 2020 | 24 May 2021 | 23 June 2021 | 8 July 2021 | 21 July 2021 | 16 May 2022 | 8 June 2022 | 13 July 2022 | 30 May 2023 | 11 July 2023 |
|---|---|---|---|---|---|---|---|---|---|---|---|---|---|---|
| Kabot Khorasan | CT | 0.61 | 0.12 | 0.58 | 0.16 | 0.1 | 0.13 | 0.09 | 0.17 | 0.66 | 0.64 | 0.31 | 0.42 | 0.92 |
| | MTC | 0.46 | 0.15 | 0.52 | 0.12 | 0.11 | 0.14 | 0.11 | 0.18 | 0.5 | 0.67 | 0.24 | 0.39 | 0.82 |
| | MTD | 0.53 | 0.17 | 0.57 | 0.15 | 0.01 | 0.15 | 0.13 | 0.16 | 0.39 | 0.62 | 0.23 | 0.52 | 0.59 |
| | CT | 0.64 | 0.08 | 0.35 | 0.09 | 0.08 | 0.16 | 0.14 | 0.19 | 0.37 | 0.44 | 0.21 | 0.49 | 0.56 |
| | MTC | 0.32 | 0.17 | 0.46 | 0.09 | 0.11 | 0.15 | 0.17 | 0.15 | 0.33 | 0.37 | 0.19 | 0.72 | 0.71 |
| | MTD | 0.18 | 0.15 | 0.43 | 0.12 | 0.12 | 0.09 | 0.24 | 0.13 | 0.23 | 0.41 | 0.19 | 0.73 | 0.58 |

where CT = conventional tillage; MTC = minimum tillage; MTD = minimalization tillage.

Coefficients of correlation (R) and determination ($R^2$) between the Crop Water Stress Index and selected spectral indices, DTM, and dates of scanning as independent variables were calculated based on forward stepwise linear regression (FSLR). The results of calculation were evaluated for individual variants and growth stages and are given in Table 6. These independent variables were selected as indicators of both the current state of the vegetation in terms of structure and vitality (NDVI), as well as the chlorophyll content or nitrogen supply with respect to the spectral bands included in the calculation (GNDVI, CIR). Since the individual parcels were located on a gentle slope, the influence of DTM also played a certain role here. In the same way, the date of scanning for individual years was an important parameter, especially during tillering, as it reflected meteorological conditions and the development of the stand in terms of phenological phases.

From the point of view of the influence of the variables on the water stress of the stands, it is clear from Table 6 that these variables had a greater influence in the ancient Khorasan variety. In general, the results in Table 6 have the opposite trend to the results in Table 5, where the assessment of the direct effect of tillage on plant water management is shown. The variables in Table 5 are then indirect indicators of the state of the stand for varieties and tillage variants.

During tillering, the date of crop scanning, which indicates the structure including crop density (NDVI), and the phenological phase reflected in the degree of greenness as an expression of chlorophyll content in leaves (GNDVI) or chlorophyll content in cellular tissues (CIR) had the greatest influence on water stress. This agrees with the study by Hoffmann et al. [58]. They found out that the NDVI best represented Leaf Area Index (LAI) measurements as an indicator of crop greenness and enabled the detection of the

developmental stage of crops in the late growing season. DTM played no role here, meaning that the stand was equally stressed regardless of location.

**Table 6.** Multiple coefficients of correlation (R) and coefficients of determination ($R^2$) between Crop Water Stress Index (CWSI) and variables of NDVI = Normalized Difference Vegetation Index; GNDVI = Green NDVI; CIR = Chlorophyl Index Red Edge; DTM = Digital Terrain Model; and Date = date of scanning resulting from forward stepwise linear regression (FSLR) for individual variants and growth stages. All coefficients are significant at 5% significance level.

| | | | | | Tillering | | | | | | |
|---|---|---|---|---|---|---|---|---|---|---|---|
| **Var** | **Mul R** | **Var** | **Mul R** | **Var** | **Mul R** | **Var** | **Mul R** | **Var** | **Mul R** | **Var** | **Mul R** |
| **Khorasan 1 CT** | | **Khorasan 2 MTC** | | **Khorasan 3 MTD** | | **Kabot 1 CT** | | **Kabot 2 MTC** | | **Kabot 3 MTD** | |
| Date | 0.31 | Date | 0.47 | Date | 0.36 | Date | 0.08 | Date | 0.18 | CIR | 0.24 |
| NDVI | 0.48 | NDVI | 0.64 | NDVI | 0.5 | CIR | 0.32 | CIR | 0.33 | GNDVI | 0.44 |
| GNDVI | 0.74 | GNDVI | 0.74 | GNDVI | 0.58 | GNDVI | 0.42 | GNDVI | 0.68 | NDVI | 0.44 |
| DTM | 0.75 | CIR | 0.76 | CIR | 0.62 | NDVI | 0.48 | NDVI | 0.68 | Date | 0.44 |
| CIR | 0.76 | DTM | 0.76 | DTM | 0.62 | DTM | 0.51 | DTM | 0.68 | DTM | 0.44 |
| **$R^2$** | **0.57** | | **0.58** | | **0.38** | | **0.26** | | **0.46** | | **0.2** |
| | | | | | **Stem Elongation** | | | | | | |
| GNDVI | 0.92 | GNDVI | 0.9 | GNDVI | 0.86 | GNDVI | 0.8 | GNDVI | 0.71 | NDVI | 0.75 |
| CIR | 0.93 | DTM | 0.91 | CIR | 0.087 | Date | 0.8 | Date | 0.73 | DTM | 0.76 |
| DTM | 0.94 | NDVI | 0.92 | DTM | 0.87 | DTM | 0.81 | DTM | 0.75 | Date | 0.77 |
| Date | 0.94 | CIR | 0.93 | Date | 0.88 | NDVI | 0.81 | CIR | 0.76 | GNDVI | 0.78 |
| NDVI | 0.94 | Date | 0.93 | NDVI | 0.88 | CIR | 0.81 | NDVI | 0.76 | CIR | 0.79 |
| **$R^2$** | **0.88** | | **0.86** | | **0.77** | | **0.65** | | **0.58** | | **0.63** |
| | | | | | **Flowering** | | | | | | |
| Date | 0.81 | NDVI | 0.82 | NDVI | 0.7 | NDVI | 0.7 | NDVI | 0.8 | NDVI | 0.81 |
| GNDVI | 0.86 | Date | 0.85 | DTM | 0.76 | DTM | 0.75 | DTM | 0.84 | Date | 0.82 |
| CIR | 0.86 | DTM | 0.87 | CIR | 0.77 | Date | 0.76 | Date | 0.86 | GNDVI | 0.83 |
| DTM | 0.87 | GNDVI | 0.88 | Date | 0.78 | CIR | 0.77 | GNDVI | 0.86 | CIR | 0.84 |
| NDVI | 0.87 | CIR | 0.88 | GNDVI | 0.79 | GNDVI | 0.77 | CIR | 0.86 | DTM | 0.84 |
| **$R^2$** | **0.76** | | **0.77** | | **0.62** | | **0.59** | | **0.75** | | **0.7** |

Var = variable; Mul R = multiple coefficients of correlation.

The stem elongation phenological phase was characterized by the detection of water stress in the canopy, which refers to a different degree of crop development within the scope of this growth phase. The DTM also had a greater influence here. During the phenological phase of flowering on all variants except Khorasan (CT), the total stand structure (NDVI) and DTM played a significant role in influencing water stress. It follows that most variants depended on the stage of flower development, because the canopy, as the top part of crops, was scanned. Feiziasl et al. [59] also concluded in their study that the NDVI as an indicator of vegetation cover had the main effect on Water Deficit Index (WDI) variation.

In general, it can be summarized that water stress was easily detectable by the monitored parameters in the growth phase of stem elongation, when the surface of the stand consists of layers of leaves or a flag leaf.

### 3.3. The Results of Intact Soil Samples

The soil bulk density graph (in Figure 4) presents the results from the three measured years from 2020 to 2022. In general, it is clear from the graph that the lowest bulk density values were relatively consistently achieved with the CT variant over the years. Again, higher values were consistently achieved over the years for the MTC variant. On the contrary, MTD technology showed fluctuations in bulk density over the years.

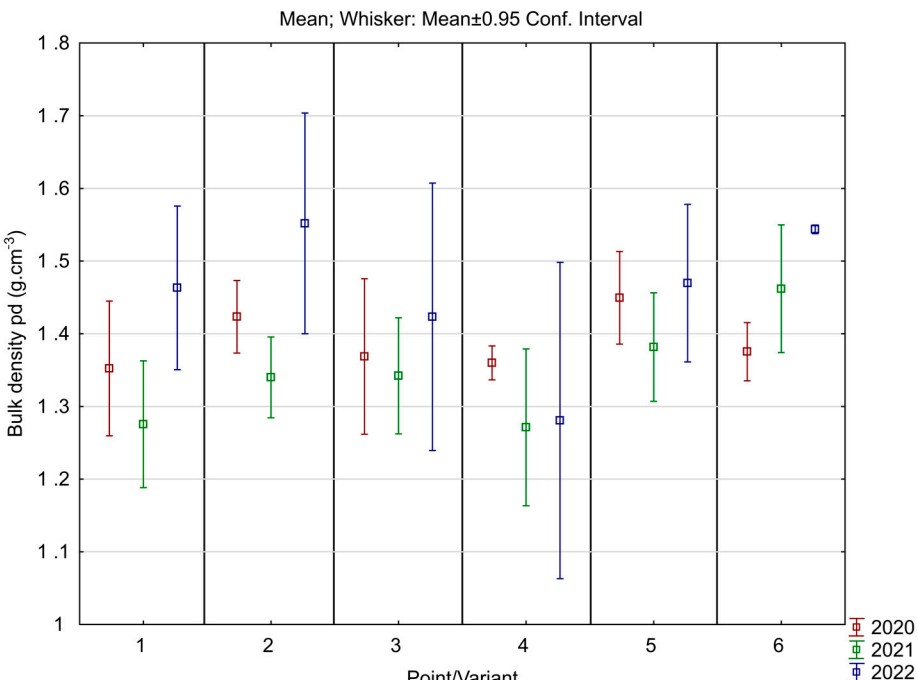

**Figure 4.** Graph of soil bulk density for the years 2020, 2021, and 2022. 1–3: Khorasan variety; 4–6: Kabot variety; 1 and 4: conventional tillage (CT); 2 and 5: minimum tillage (MTC); 3 and 6: minimization tillage (MTD).

All years showed interesting results in the ploughed variants. In the first year, the CT variants had the lowest values of bulk density, followed by MTD with almost the same average values, and the highest values of bulk density were measured for MTC tillage. The second year (2021) differed in the swapping order of the MTD and MTC variants. This means that the order from lowest to highest in the second year was CT, MTC, and MTD, where the variant sown with the Kabot variety with MTD tillage showed significantly higher values than other variants, with a statistically significant difference compared to the CT variants.

In the last measured year, 2022, the variant sown with Khorasan wheat copied the or-der of the year 2021, where the lowest values were for CT followed by MTD and the highest value of bulk density was for MTC. In contrast to the second variant sown with Kabot wheat, the MTD tillage system performed the worst. However, the best results were achieved mainly in ploughing, which agrees with most authors [60–62]. On the other hand, variants 3 and 6 (MTD—Khorasan and Kabot) were almost identical to CT in terms of bulk density in the first year of the experiment. This could suggest that under certain conditions (such as soil type, temperature, precipitation, crop, etc.), some no-tillage technologies can achieve the same [16], and perhaps in some cases better, results than CT [63]. This was also stated in the publication by Woźniak and Rachoń [52], where suitable soil conditions for plant growth were created by CT mainly on medium-moist soils, while on dry and semi-arid soils, it was better to choose systems without plowing. A graph of the total porosity from the experimental plots can be seen in Figure 5. The porosity results were in good agreement with the bulk density results during the monitored years. In years when greater bulk density was measured, porosity was measured to be lower and vice versa.

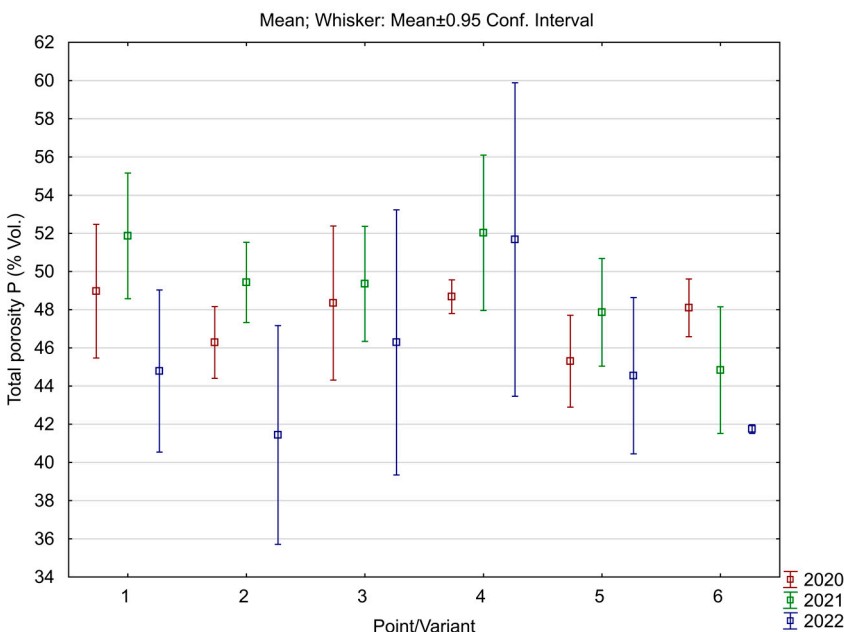

**Figure 5.** Graph of total porosity for the years 2020, 2021, and 2022. 1–3: Khorasan variety; 4–6: Kabot variety; 1 and 4: conventional tillage (CT); 2 and 5: minimum tillage (MTC); 3 and 6: minimization tillage (MTD).

As expected, the results mirrored the bulk density of the soil, which means that the CT variants showed the highest proportion of pores in the first and second years (2020 and 2021) of measurement. According to a study by Mehra et al. [51], soil porosity was greater in a CT system than in a no-till system; moreover, the macropore content was dominant in the CT system, which is in line with our study. In 2022, the MTC tillage system for the variant sown with Khorasan wheat had a higher pore content than CT, while the variant sown with the modern Kabot wheat variety on an MTC system showed the lowest pore content.

The total porosity and bulk density results were consistent with the CWSI results (Table 5) and are consistent with the following statements. According to Woźniak and Rachoń [52], greater pore content and soil compaction in CT indicated higher soil loosening and resulted in greater soil water loss. Plants that did not have as much available water responded by changing leaf size or reducing stem elongation and stomatal closure to reduce evapotranspiration, increasing their stress [54]. According to Luan and Vico [64], water stress resulted in an increase in leaf/canopy temperature (Tc). This is also stated in research by James et al. [65], where transpiration was an important factor influencing the overall plant temperature during water stress. Liebhard et al. [66] stated that it depends on soil cultivation, which affects the development of plants and their roots, and at the same time affects water intake and evapotranspiration itself. Different tillage methods have a key influence on the gradual development, vitality, and yield of plants as they are closely related to local climatic and soil conditions, as was presented in the study by Blanco-Canqui and Wortmann [67].

## 4. Conclusions

The results showed that the method of tillage plays a significant role for the selected wheat varieties Khorasan and Kabot, as it affects their growth and vitality during the growing season. The condition of the stands was monitored using a UAV and selected calculated indices (NDVI, GNDVI, CSWI, and CIR). It is clearly seen from the results that in Khorasan wheat, all values were lower in BBCH tillering than in Kabot wheat.

The dry years affected both varieties, especially the content of chlorophyll and crop structure. The values of the calculated spectral indices were lower than in years with

higher rainfall totals. In addition, the CT variant caused the worst retention capacity in the soil, which was reflected in the CWSI results, when the plowed variants performed the worst, while the MTD variants performed the best. The results of bulk density and total porosity were consistent with the results of the NDVI and CWSI. Due to greater evaporation from the soil in the CT variant, there was less plant growth and a higher degree of stress due to stomatal closure. The results showed that crops can be effectively monitored during the growing seasons with the help of selected indices, and it is then possible to react flexibly to changes, deficiencies, or other problems. Regarding the individual methods of tillage and water stress, the ancient variety Khorasan performed the worst on the CT variant. MTD appeared to be the best tillage method for Khorasan cultivation in terms of water management.

Based on our results from plot trials, research on the behavior of ancient varieties depending on different tillage methods over larger areas and in different soils can be recommended.

**Author Contributions:** Conceptualization—K.B. and J.K.; methodology—K.B. and J.K.; software—J.K.; validation—J.C.; formal analysis—J.C. and J.K.; investigation—K.B., J.C. and J.K.; resources—K.B., J.C. and J.K.; data curation—J.K.; writing—original draft preparation—J.K., K.B. and J.C.; writing—review and editing—J.K., K.B. and J.C.; visualization—J.K. and J.C.; supervision—J.K. All authors have read and agreed to the published version of the manuscript.

**Funding:** This paper was financed by an internal project of the Faculty of Engineering of the Czech University of Life Sciences Prague No.: 2020: 31160/1312/3110; 2022: 31160/1312/3101.

**Data Availability Statement:** Data are contained within the article.

**Conflicts of Interest:** The authors declare no conflicts of interest.

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
