# Peer review of "Unmanned-Aerial-Vehicle Data as an Effective Tool for the Evaluation of Ancient Khorasan and Modern Kabot Spring Wheat Varieties under Different Tillage Systems"

_agronomy, doi:10.3390/agronomy14010147_

Round 1

Reviewer 1 Report

Comments and Suggestions for Authors

1. It is necessary to describe the graphical dependencies presented in Figures 2-4 indicating changes in the quantitative values of the parameters and characteristics under study.

2. It is necessary to reflect the results of statistical processing of the obtained results of experimental studies.

3. It is necessary to present a graphical dependence of the data presented in Table 2 and reflect their distribution law during the accounting period.

4. The instruments and equipment used in conducting experimental studies, as well as their technical characteristics, should be reflected.

5. It is necessary to indicate the prospects for further implementation of the research results obtained.

Reviewer 2 Report

Comments and Suggestions for Authors

The reviewed article, titled "Remote Sensing Data as An Effective Tool for Evaluation of Ancient Khorasan and Modern Kabot Spring Wheat Varieties under Different Tillage”, is a research study showcasing the application of remote sensing in agronomy. It primarily assesses how various wheat varieties respond to meteorological conditions by utilizing remote sensing data, particularly UAV imagery, to extract spectral indices, including a water stress index. The goal is to evaluate their capacity to withstand drought and high temperatures, considering diverse tillage techniques, namely conventional, minimum, and minimalization tillage methods. A more in-depth and detailed examination of the manuscript reveals specific remarks, particularly:

1)     The authors have employed UAV imagery in their study. I suggest considering its inclusion in the title instead of using the term "remote sensing data."

2)     Line 13: When evaluating crops, it is common to concentrate on specific parameters or indicators, such as biomass. I recommend that the authors explicitly mention these in the abstract for improved clarity and understanding.

3)     Line 21: The abbreviation CWSI is not mentioned in the abstract.

4)     Line 30: Provide a reference for this sentence: “wheat is the second largest grain worldwide”.

5)     Line 41: The authors can use this sentence: "as pointed out by research conducted by Nakforosh et al. [8]." instead of: "as research points out Nakforosh et al. [8]."

6)     Line 41: In the reference list, the author’s name is “Nakhforoosh” not “Nakforosh”. The authors should ensure accuracy in referencing.

7)     Line 93: I suggest that the authors add the term "methods or techniques" to the sentence "The most common tillage," so that readers can clearly understand what the authors are presenting in this subsection.

8)     After introducing the tillage variants in the introduction section, the authors may consider adding a table summarizing the three different types to enhance the readability of this subsection.

9)     Line 125: The authors should use the present tense when discussing the objectives of this study. They can use: "The objective of this study is to assess Khorasan and Kabot spring wheat.

10) In the introduction section, relevant studies related to this research should be presented, and their limitations should be indicated.

11) The authors should clearly state their contribution in this research study in both the abstract and the introduction section.

12)  Line 126: The authors should specify the four considered growing seasons.

13) Line 163: The authors should provide the meaning of NPK.

14) Line 172: I think when the authors mention four growing seasons, they are referring to the period from 2020 to 2023. However, for improved reader comprehension, they can specify the four-year span.

15) Line 179: Why did you select the specified growth stages outlined in Table 2, and what accounts for the variations between the stages from one year to another?

16) When presenting the methodological section, it is preferable to use a diagram that structures key points from the methodology.

17) Line 188: Lateral overlap, longitudinal overlap, flight height, and spatial resolution should not be presented in the first column of Table 3 since they are not a sensor or a camera.

18) Line 201: The column labeled "algorithm" should be renamed to "equation" or "expression," and the column "Developed by" should be renamed to "used for."

19) Line 201: The digital terrain model is not a spectral index; it should be removed from this table.

20) Line 215: The authors should rewrite the idea as there are some grammatical issues, and the sentence is too lengthy.

21) Why did you segregate the crop water stress index from the other spectral indices when presenting the results?

22) In their conclusion, the authors should indicate future directions and perspectives for work that can be considered based on their results.

In general, the manuscript should indicate other methods or traditional techniques of evaluation for the presented varieties. The proposed methodology should be described in a way that addresses existing issues and highlights the introduced novelties. Based on all the remarks above, my recommendation is to accept with major revision.
